# Rolling Circle Amplification-Enabled Ultrasensitive Point-of-Care Test Method for Aflatoxin B1 in the Environment and Food

**DOI:** 10.3390/foods13193188

**Published:** 2024-10-07

**Authors:** Hongyu Duan, Yuan Zhao, Xiaofeng Hu, Meijuan Liang, Xianglong Yang, Li Yu, Behrouz Tajdar Oranj, Valentin Romanovski, Peiwu Li, Zhaowei Zhang

**Affiliations:** 1Key Laboratory of Biology and Genetic Improvement of Oil Crops, Hubei Hongshan Lab, Oil Crops Research Institute of Chinese Academy of Agricultural Sciences, Wuhan 430062, China; dhy2634591222@163.com (H.D.); zhao_yuan2494@126.com (Y.Z.); xiaofenghu85@126.com (X.H.); liangmj@whu.edu.cn (M.L.); xianglongyang@foxmail.com (X.Y.); yuli01@caas.cn (L.Y.); peiwuli@oilcrops.cn (P.L.); 2Research Center for Environmental Determinants of Health (RCEDH), Kermanshah University of Medical Sciences, Kermanshah 67146, Iran; tajdar.tums@yahoo.com; 3Center of Functional Nano-Ceramics, National University of Science and Technology MISIS, Moscow 101000, Russia; vramano@kth.se

**Keywords:** aflatoxin, rolling circle amplification, environmental monitoring, food safety

## Abstract

Aflatoxin B1 (AFB1) contamination poses a fatal risk to human beings and urgently needs highly sensitive detection for environmental monitoring and food safety. However, the existing challenges are the unsatisfied sensitivity of the immunoassay methods and the complex matrix effect. Rolling circle amplification (RCA) is a promising method for nucleic acid isothermal amplification due to its high specificity and sensitivity. Herein, we constructed a general RCA-based point-of-care test method (RCA−POCT). With biotinylated antibodies, streptavidin, and biotinylated RCA primers, we realized the signal transduction and preliminary signal amplification. In this way, the fluorescent signal of the immunocomplex on the microwells was greatly enhanced. Under optimal conditions, we recorded sensitive detection limits for aflatoxin B1 (AFB1) of 1.94, 16.3, and 37.7 fg/mL (femtogram per microliter), and wide linear ranges with 5 × 10^−6^ to 5, 5 × 10^−5^ to 5, and 5 × 10^−5^ to 5 ng/mL in the irrigation water, field soil, and peanut samples, respectively. Satisfactory recovery, specificity, repeatability, and reproducibility were observed. The RCA−POCT was validated by comparing it to the HPLC method. This work provides a general RCA-assisted detection method for AFB1 in the environment and food.

## 1. Introduction

The isothermal amplification method allows for rapid and efficient nucleic acid amplification, avoiding the thermal cycles of the traditional polymerase chain reaction (PCR) [1,2]. By eliminating the need for bulky and expensive equipment, isothermal amplification mitigates the risk of temperature-sensitive biomolecule loss in the reaction progress. Among the various isothermal amplification techniques, rolling circle amplification (RCA) with a single enzyme and a single primer can amplify a single copy to a readily detectable level [3]. Therefore, RCA, as a simple amplification technique, is extensively used not just for biological assays, but also to improve the sensitivity, selectivity, and detection range of the assays. For instance, RCA can be easily integrated with portable devices, making it applicable in biosensor development [4], such as suspension bead arrays [5], lab on paper [6], microfluidic chips [7], and lateral flow assay [8]. The research into these applications mainly focused on the protein, DNA, or RNA targets, ignoring the detection dilemma of small molecules.

Aptamers, for example, can recognize the small molecules with specificity and enable direct signal conversion and amplification in mycotoxin-detection assays in food matrices [9,10]. However, the rarity of available aptamers hampers their wide application in point-of-care testing (POCT) in real samples because aptamer affinity is strongly dependent on its binding condition. Monoclonal antibodies serve as highly specialized recognition tools for small molecules, showing robust characterization, especially in laboratory and commercial applications. Small molecular immunorecognition detection assays confront the challenges of effective signal amplification approaches and complex sample matrix effects. To improve the practical application of signal transferring, our group previously developed a robust antibody-enabled POCT with a CRISPR/Cas12a system for small molecules [11]. This method realized signal transferring from antibody–antigen to nucleic acid by strongly non-covalent binding of biotin and streptavidin [12].

As a class I carcinogen [13], AFB1 is transferred from the environment to food through a chain of irrigation: water to field and soil to peanut [14]. The current immunological test methods for AFB1 detection are based on colorimetry [15], electrochemiluminescence [16], electrochemistry [17], lateral flow fluorescent strip [18], fluorescence resonance energy transfer [19], and photoelectrochemistry [20]. Nevertheless, these above methods lacked sensitivity to the complex matrix effects. Harnessing the simplicity and specificity of the RCA nucleic-acid-detection method, we proposed an RCA−POCT method to combine the streptavidin-biotin system and RCA to boost sensitivity in AFB1 detection in complicated matrices. We fabricated a biotinylated antibody-streptavidin-biotinylated single-stranded DNA (biotin-ssDNA) sandwich to initiate RCA for AFB1 detection in real samples. The above-mentioned sandwich complexes enabled simultaneous immune recognition of antigens and signal transformation. Then, we used RCA to amplify biotin-ssDNA to a long-chain ssDNA expediently. The long-chain ssDNA was partially complementary to plenty of fluorescently labeled signal probes, causing strong fluorescence signal production. The fluorescence intensity was recorded using a microplate reader. After optimization, the irrigation water, field soil, and peanut were used to evaluate the linear range, limit of detection, recovery, specificity, repeatability, and reproducibility of this method. Validation was conducted by comparing results via the RCA−POCT to those via the HPLC method. This RCA−POCT with high sensitivity and selectivity promises potential in environmental monitoring and food safety.

## 2. Materials and Methods

### 2.1. Regents and Instruments

DNA sequences [21] (Table 1) were synthesized by Sangon Biotech (Shanghai, China).

All the reagents and instruments used are listed in the following. Standard aflatoxin B1 (AFB1), ochratoxin A (OTA), deoxynivalenol (DON), zearalenone (ZEN), fumonisin B1 (FB1), sterigmatocystin (ST), diacetoxyscirpenol (DAS) solution, bovine serum albumin (BSA), AFB1-BSA, streptavidin, and 0.22 μm membrane filters were purchased from Sigma-Aldrich Co., Ltd. (Rocklin, CA, USA). Anti-AFB1 monoclonal antibody (Anti-AFB1 mAb) was prepared in our lab. Exonuclease I (Exo I, 20 units/μL), Exonuclease III (Exo III, 100 units/μL), and 10 × NEBuffer™ 1 were obtained from New England Biolabs (Ipswich, UK). DNA marker (20 bp DNA Ladder) and 6 × DNA loading buffer were ordered from Takara Biomedical Technology (Beijing, China). Red fluorescent nucleic acid dye (10,000 × SolarGelRed), and nuclease-free water were purchased from Beijing Solarbio Science & Technology Co., Ltd. (Beijing, China). The native polyacrylamide gel electrophoresis (PAGE) kit was purchased from Beyotime Biotech (Beijing, China). T4 DNA ligase, 10 × T4 DNA ligase buffer, deoxyribonucleoside (dNTP, 10 mM), phi29 DNA polymerase, 10 × phi29 buffer, tris-borate-EDTA (TBE) buffer, and biotin conjugation kit (D601048) were acquired from Sangon Biotech (Shanghai, China). The 96-well microplates utilized were bought from Corning Inc. (New York, NY, USA). Water used throughout this work was purified with a Milli-Q system (Millipore, Billerica, MA, USA). The AFB1-BSA was diluted with carbonate buffer (0.05 mol/L, pH = 9.6), which was prepared by adding 0.795 g Na_2_CO_3_ and 1.465 g NaHCO_3_ to ultrapure water to 500 mL. The 70% methanol solution was prepared by adding methanol to ultrapure water solution (70:30, *v*/*v*). PBS (0.1 mol/L, pH = 7.4) was prepared with 24.0 g NaCl, 8.7 g Na_2_HPO_4_·12H_2_O, 0.2 g KCl, and 0.2 g KH_2_PO_4_ added to ultrapure water to 1000 mL. The fluorescence intensity was recorded using a SpectraMax i3x Microplate Reader from Molecular Devices (Sunnyvale, CA, USA) at the excitation wavelength (498 nm) and emission wavelength (512 nm). Peanuts, peanut soil, and irrigation water were collected from our laboratory in 2022.

### 2.2. Preparation of Biotinylated Anti-AFB1 mAbs

The biotinylated anti-AFB1 monoclonal antibodies (anti-AFB1 mAbs) were prepared using a commercial biotin conjugation kit (see Appendix A). The below steps were performed according to the instructions provided with the kit. Briefly, the homemade anti-AFB1 mAb (500 μL), activated biotin (250 μL), and biotin conjugation solution (250 μL) were mixed for 2 h and purified using a desalting column (provided in the kit). The biotinylated anti-AFB1 mAbs were obtained and kept at −20 °C before use.

### 2.3. Synthesis of Circular DNA

The primer and padlock with the same molar concentration were added to 20 μL of 1 × T4 DNA ligase buffer to anneal at 95 °C for 5 min and then 25 °C for 30 min. The above solution was mixed with 2 μL of T4 DNA ligase overnight at 25 °C. We added Exo I (1 μL), Exo III (0.5 μL), 10 × Exo I buffer (5 μL), and 10 × NEBuffer™ 1 (5 μL) and adjusted to 50 μL with nuclease-free water. We obtained circular DNA by incubation for 1 h at 37 °C. The cyclization of the padlock was characterized using 12% native polyacrylamide gel electrophoresis (PAGE) analysis.

### 2.4. Procedure of the RCA−POCT

We verified the feasibility of the RCA reaction before developing the RCA−POCT procedure. A microwell was washed 3 times with PBST between each step. After being pretreated with AFB1 antigen and blocked with BSA, we added AFB1 and biotinylated anti-AFB1 mAb and incubated at 37 °C for 45 min. Then, we added 100 µL streptavidin (0, 1, 5, 10, 15, and 20 μg/mL) in the microwell and incubated at 37 °C for 30 min. The 50 µL of biotinylated primer (0, 0.48, 0.72, 0.96, 1.2, and 1.44 μM) was added and incubated for 30 min at 37 °C. After that, we added a 50 μL RCA reaction system, containing circular DNA (0, 0.3, 0.6, 0.9, 1.2, and 1.5 μM), 1 mM dNTPs, phi29 DNA polymerase (0, 0.1, 0.2, 0.3, 0.4, and 0.5 units/μL), and 0.2 mg/mL BSA and 1 × phi29 buffer. The RCA reaction was conducted at 37 °C for 0, 1.5, 2, 2.5, 3, and 3.5 h, respectively. Next, 50 µL of 5-carboxyfluorescein (FAM) labeled signal probe (0, 0.1, 0.5, 1, 1.5, and 2 μM) was added and incubated for 30 min. After adding 100 μL PBS, the fluorescence intensity was measured with a microplate reader. All experiments were repeated at least three times.

### 2.5. Evaluation of the RCA−POCT

The sensitivity, specificity, repeatability, and reproducibility of the RCA−POCT were evaluated by spiking experiments in irrigation water, field soil, and peanut samples, respectively. The calibration curve was established as Y = aX + b. Y was the fluorescence intensity, and X was the logarithm of AFB1 concentration. LOD = $\bar{X}$ +3δ was calculated with 20 blank samples. δ was the standard deviation of testing results of AFB1 in 20 blank samples and $\bar{X}$ was the average AFB1 concentration in 20 blank samples [22].

To evaluate the specificity of the RCA−POCT, seven co-existing mycotoxins, including AFB1, ochratoxin A (OTA), deoxynivalenol (DON), zearalenone (ZEN), fumonisin B1 (FB1), sterigmatocystin (ST), and diacetoxyscirpenol (DAS) were tested at a concentration of 100 ng/mL. Repeatability and reproducibility were assessed by spiking AFB1 in three samples with final concentrations of 2, 5, and 10 ng/mL, respectively.

### 2.6. Validation of the RCA−POCT with HPLC in Real Samples

To verify the RCA−POCT in real environmental and food samples, we compared the results between the RCA−POCT and HPLC methods in irrigation water, field soil, and peanut (Appendix A).

### 2.7. Sample Preparation

Modified sample treatment was performed according to our previous work [22,23]. For irrigation water, the liquid sample was filtered using a 0.45 μm hydrophilic filter. For field soil and peanuts, the samples were crushed and sieved to 2 mm. A 5 g sample in 20 mL 70% methanol solution was agitated in a shaker for 20 min and then centrifuged at 8084× *g* for 10 min. The supernatant was filtered using a filter (0.22 μm). The filtrate was used for HPLC detection directly or was diluted 7-fold with PBS for the RCA−POCT.

## 3. Results and Discussion

### 3.1. Principle of the RCA−POCT

The principle of the RCA−POCT is detailed in Figure 1. First, the antigen AFB1-BSA and BSA solution were used to coat and block the microwell surface, respectively, followed by adding the test sample solution and biotinylated anti-AFB1 mAbs solution. In negative samples, the biotinylated anti-AFB1 mAbs were bound to antigen AFB1-BSA. The immunoassay signal was then transferred and amplified to the biotinylated primer through the biotin-streptavidin system. Next, the conjugated biotinylated primer was served to initiate the RCA reaction. The RCA reaction involved three major steps. (1) Circular DNA formation of biotinylated primers and padlocks in the action of DNA ligase. (2) Circular DNA purification with exonuclease I and exonuclease III. (3) The RCA reaction initiation in the presence of dNTPs and Phi29 polymerases. Long ssDNAs with tandem repeating sequences complementary to the circular DNA are produced, each of which can be hybridized with FAM-labeled signal probes. Therefore, the long-stranded RCA products attached to the microwells could be analyzed based on the FAM fluorescence signal. The fluorescence intensity was measured through a microplate reader with a negative correlation between the concentration of AFB1 in the samples and the fluorescence intensity.

### 3.2. The Feasibility of the RCA−POCT

The feasibility of the RCA−POCT was confirmed by utilizing 12% native PAGE electrophoresis (Figure 2). As shown in Figure 2, the DNA ladder was used to indicate the position of dsDNA of different molecular weights, while the PAGE was used as a tool for separating ssDNA. Therefore, the DNA ladder as a whole was positioned close to the loading hole. This explains that the primer (56 nt) position of Lane 1 was closer to the 40 bp position of the DNA ladder. The circular DNA (Lane 3) trailed behind the padlock (Lane 2), proving the synthesis of circular DNA. Due to the difference in DNA conformation, circular DNA shifted at a slower rate than linear DNA in 12% native PAGE electrophoresis, and therefore circular DNA was positioned higher in 12% native PAGE electrophoresis.

The high molecular weight product of primer amplification was detected in the sample loading hole in Lanes 5–7, and the quantities of RCA products increased with the amplification time. Additionally, the disappearance of the primer and padlock band (Lane 4) confirmed the consumption of primer by the RCA products. To validate the application of the RCA−POCT in the microwell, a compatibility test was conducted to prove the validity of the RCA reaction in a preloaded streptavidin microwell with solid support. The presence of biotinylated primers (0.625 µM) significantly increased the fluorescence intensity as compared to its absence (Appendix A). Similarly, the fluorescence intensity increased rapidly in the preloaded AFB1-BSA microwell with the presence of biotinylated anti-AFB1 mAb (Appendix A). The immunoassay signal was transmitted to the biotinylated primers before being expressed as fluorescence signals.

### 3.3. Optimization of the RCA−POCT

To improve the analytical performance of the proposed RCA−POCT, we initially optimized the concentration of streptavidin and biotinylated primer via the intensity of fluorescence generated by biotin and FAM double-labeled primer in the pre-loaded microwell. The RCA−POCT sensitivity was proportional to the concentration of streptavidin and biotinylated primer. With an increase in streptavidin dosage, fluorescence intensity improved rapidly up to 15 µg/mL, after which there was little change in intensity up to 20 µg/mL (Figure 3A), indicating that 15 µg/mL streptavidin was sufficient for immobilizing the biotinylated primer. Similarly, the fluorescence intensity increased with biotinylated primer, with the highest fluorescence intensity achieved at 1.2 µM (Figure 3B).

To ensure that RCA products reach sufficient lengths, thus maximizing signal probe labeling, we investigated the optimal concentrations of circular DNA and phi29 DNA polymerase. As the concentration of circular DNA and phi29 DNA polymerase increased, fluorescence intensities increased rapidly. The intensities for circular DNA and phi29 DNA polymerase reached a plateau at 1.2 µM and 0.4 U/µL, respectively (Figure 3C,D). We analyzed the effect of RCA reaction time on fluorescence intensity using a predetermined amount of circular DNA and phi29 DNA polymerase. During the detection process, it was determined that a minimum of three hours was required for signal saturation for later experiments (Figure 3E).

The concentration of the FAM-labelled signal probe was optimized, as it directly influences fluorescence intensity. An increase in signal probes led to heightened sensitivity, though excessive probes resulted in non-specific adsorption. An increase in signal probe concentration caused a significant increase in fluorescence intensity, eventually plateauing at 1.5 µM (Figure 3F). This plateauing was likely due to the spatial steric effect and limited space. Thus, 1.5 µM signal probes were used in this assay.

### 3.4. Analytical Performance of the RCA−POCT

The study aimed to evaluate the sensitivity of the RCA−POCT for quantifying AFB1 under optimal conditions. AFB1 detection was performed as described in Section 2.5 in the irrigation water, field soil, and peanut samples. Various concentrations of AFB1 were added to negative irrigation water, field soil, and peanut samples, and the fluorescence intensity was measured. There was a linear relationship between the logarithm of AFB1 concentration and fluorescence intensity in irrigation water samples in the range of 5 × 10^−6^ to 5 ng/mL (Figure 4A), field soil samples in the range of 5 × 10^−5^ to 5 ng/mL (Appendix A), and peanut samples in the range of 5 × 10^−5^ to 5 ng/mL (Appendix A), respectively. The LOD was calculated according to Section 2.5 to be 1.94 fg/mL, 16.3 fg/mL, and 37.7 fg/mL in the irrigation water samples, peanut soil samples, and peanut samples, respectively. Compared with previously reported aptasensors or immunosensors for AFB1, listed in Table 2, this work exhibited obvious advantages. From the perspective of fluorescence detection methods [24,25,26,27], the LODs and linear ranges of aptamer-based fluorescence detection were inferior to that of the RCA−POCT method. Further, from the point of view of the POCT, the LODs using smartphones [22,28] as the final signal output are higher than the fluorescence-based method. Carbon quantum dot immunosensor, multimode nanozyme-linked immunosorbent assay, and this RCA−POCT method showed similar linear ranges, but this RCA−POCT was more suitable for trace AFB1 and complex matrix.

The specificity of the RCA−POCT was evaluated by spiking experiments to determine interferences from co-existing mycotoxins, including AFB1, OTA, DON, ZEN, FB1, ST, and DAS. Our laboratory has previously confirmed the high sensitivity and high specificity of anti-AFB1 monoclonal antibodies [32]. The results showed that fluorescence intensities were comparable to the blank for DON, OTA, ZEN, DAS, FB1, and ST. The presence of AFB1 in test samples led to a decrease in fluorescence intensity (Figure 4B). The results indicated that the RCA−POCT has high specificity in irrigation water, field soil, and peanut matrices.

The repeatability and reproducibility of the RCA−POCT were evaluated by recovery studies that involved the spiking of varying levels of AFB1 concentrations in blank samples. For intra-assay accuracy, recoveries were 96.7–102.5%, 92.5–99.4%, and 99.0–103.6% in irrigation water, field soil, and peanut samples, respectively. For inter-assay precision, recoveries ranged from 96.6–102.7%, 96.8–102.6%, and 91.3–103.3%, with coefficients of variation (CVs) ranging from 3.5–8.2% (Table 3). Overall, the variances detected in both intra- and inter-assay demonstrated acceptable levels for AFB1 quantitation. These results demonstrated that the RCA−POCT has good repeatability and reproducibility.

### 3.5. Validation of the RCA−POCT with HPLC in Real Sample

To evaluate the feasibility and effectiveness of the RCA−POCT for AFB1 monitoring, the study analyzed 12 irrigation water samples, 12 field soil samples, and 12 peanut samples, using both the RCA−POCT and the HPLC method. Both methods produced consistent results with a strong correlation (R^2^ = 0.992, Figure 4C), confirming the reliability, practicality, and convenience of the proposed RCA−POCT for AFB1 monitoring in the environment and food.

## 4. Conclusions

We developed an ultrasensitive RCA−POCT platform to detect AFB1 in irrigation water, field soil, and peanut samples. The RCA−POCT showed satisfactory recovery, a wide linear range, high sensitivity, repeatability, reproducibility, and specificity, making it suitable for complex matrices from food and the environment. Validation results demonstrated that the proposed method is highly practicable. Our proposed method could be utilized to screen other risk factors and may be useful for achieving the need for ultrasensitive levels in environmental monitoring, agricultural products, and foods.

## Figures and Tables

**Figure 1 foods-13-03188-f001:**
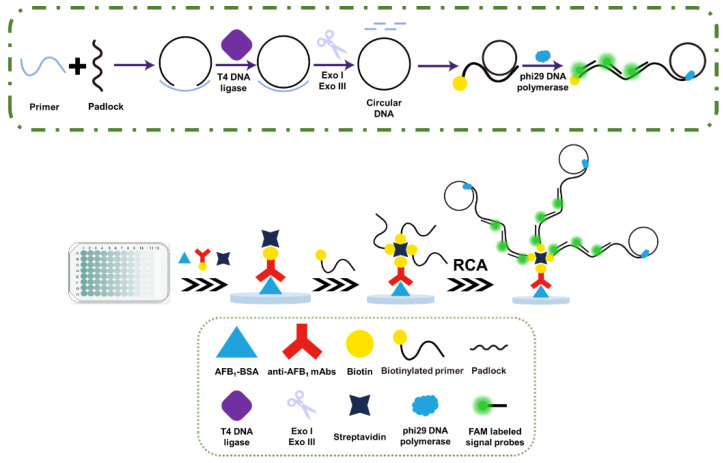
Schematic diagram of the principle of the RCA−POCT.

**Figure 2 foods-13-03188-f002:**
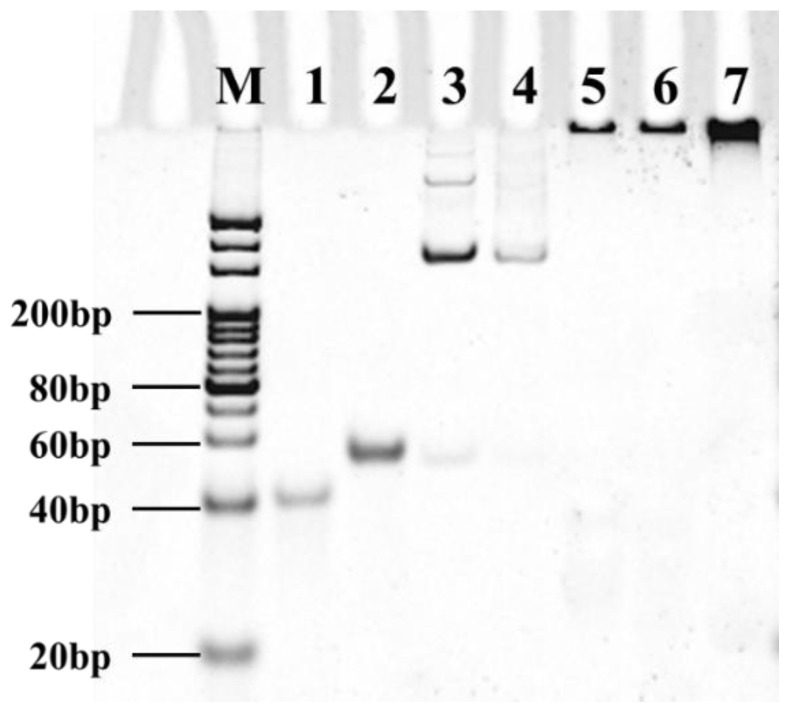
Polyacrylamide gel electrophoresis of the DNA molecules and products. (M: DNA ladder; Lane 1—primer; Lane 2—padlock; Lane 3—circular DNA; Lane 4—the hybridization of primer and circular DNA; Lanes 5–7—RCA products with 1 h, 2 h, and 3 h amplification time).

**Figure 3 foods-13-03188-f003:**
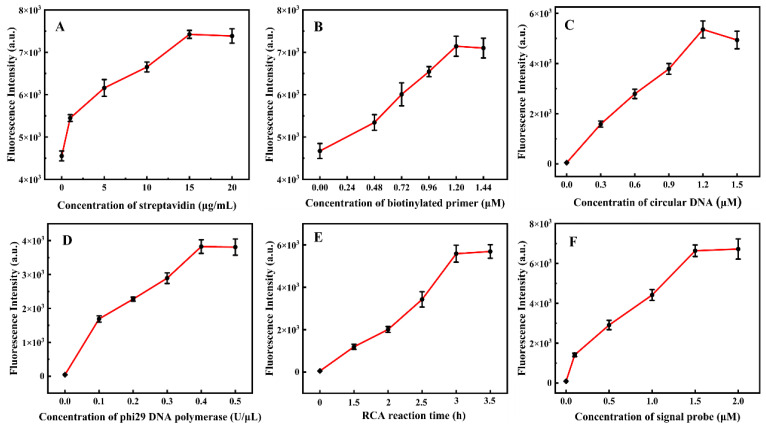
Optimization of RCA. (**A**) Influence of the concentration of streptavidin (0, 1, 5, 10, 15, and 20 µg/mL). (**B**) Influence of the concentration of primer (0, 0.48, 0.72, 0.96, 1.2, and 1.44 µM). (**C**) Influence of the concentration of circular DNA (0, 0.3, 0.6, 0.9, 1.2, and 1.5 µM). (**D**) Influence of the concentration of phi29 DNA polymerase (0, 0.1, 0.2, 0.3, 0.4, and 0.5 units/μL). (**E**) Influence of the RCA reaction time (0, 1.5, 2.0, 2.5, 3.0, and 3.5 h). (**F**) Influence of the concentration of signal probe (0, 0.1, 0.5, 1.0, 1.5, and 2.0 µM).

**Figure 4 foods-13-03188-f004:**
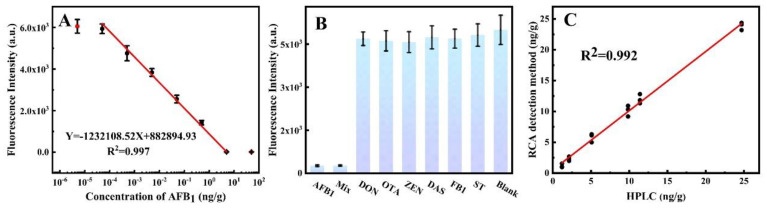
The detection of AFB1 with the RCA−POCT. (**A**) The calibration curves of AFB1 in irrigation water samples. (**B**) Selectivity tests for the detection of AFB1 over other toxins. (**C**) Correlation between AFB1 analysis data from 36 samples using HPLC (*x*−axis) and the RCA−POCT (*y*−axis).

**Table 1 foods-13-03188-t001:** DNA sequences used in this work for RCA reaction.

Name	Sequence (5′-3′)
Biotinylated primer	Biotin-GATCGGGTGTGGGTGGCGTAAAGGGAGCA TCGGACAGGCGAAGACAGGTGCTTAGT
Padlock	Phosphate-TGTCTTCGCCTGTCCGATGCTCTTCCTT GAAACTTCTTCCTTTCTTTCGACTAAGCACC
Double labeled primer	Biotin-GATCGGGTGTGGGTGGCGTAAAGGGAGCA TCGGACAGGCGAAGACAGGTGCTTAGT-FAM
Signal probe	AACTTCTTCCTTTCTTTCGACTAAGCACC-FAM

**Table 2 foods-13-03188-t002:** Comparison of this proposal with previous methods for AFB1 detection.

Method	Recognition Element	Signal Detection	LOD (fg/mL)	Linear Range (ng/mL)	References
Dual-DNA tweezers	Aptamer	Fluorescence	35,000	0.08–10	[24]
Self-assembly DNA tetrahedron	Aptamer	Fluorescence	10,000	0.05–100	[25]
RT-qPCR	Aptamer	Fluorescence	25	5 × 10^−5^–5	[26]
Fluorescent biosensor	Antibody and aptamer	Fluorescence	8.38	0.1–100	[27]
AIEgens nanosphere-POCT	Antibody	Fluorescence	3000	0.05–1.2	[22]
Colorimetric-POCT	Antibody	Colorimetry	33	0.1–50	[28]
Histidine-modified Fe_3_O_4_ nanozyme colorimetry	Antibody	Colorimetry	34	1 × 10^−4^–1	[29]
Multimode nanozyme-linked immunosorbent assay	Antibody and aptamer	Photothermal Colorimetry Fluorescence	0.54	10^−5^–100	[30]
Carbon quantum dot immunosensor	Antibody	Electrochemiluminescence	9.55	1 × 10^−4^–100	[31]
RCA−POCT	Antibody	Fluorescence	1.94	5 × 10^−6^–5	The current study

**Table 3 foods-13-03188-t003:** Accuracy and precision of the assay in AFB1-spiked samples.

Sample	Spiked (ng/g)	Intra-Assay (*n* = 3)			Inter-Assay (*n* = 3)		
		Found (ng/g)	Recovery (%)	CV (%)	Found (ng/g)	Recovery (%)	CV (%)
Peanut	2	2.03	101.5	3.5	2.07	103.3	5.5
	5	5.18	103.6	3.5	4.56	91.3	5.8
	10	9.90	99.0	4.0	9.73	97.3	7.0
Field soil	2	1.85	92.5	4.2	1.94	96.8	6.6
	5	4.97	99.4	5.6	5.13	102.6	7.2
	10	9.59	95.9	4.4	10.23	102.3	6.1
Irrigation water	2	2.05	102.5	4.4	2.05	102.7	8.2
	5	5.09	101.8	4.3	4.83	96.6	6.9
	10	9.67	96.7	4.7	9.75	97.5	6.6

## Data Availability

The data presented in this study are available on request from the corresponding author due to privacy.

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
