# Peer review of "Rolling Circle Amplification-Enabled Ultrasensitive Point-of-Care Test Method for Aflatoxin B1 in the Environment and Food"

_foods, 2024, doi:10.3390/foods13193188_

Round 1

Reviewer 1 Report (New Reviewer)

Comments and Suggestions for Authors

In this work, authors report on the development of an ultrasensitive Rolling Cycle Amplification-POCT platform to detect AFB1 in water, field soil, and food samples. The RCA-POCT sensor showed satisfactory recovery, a wide linear range, high sensitivity, repeatability, reproducibility, and specificity. This is an interesting well written paper and the results are supported by the data presented. I have the following comments for the authors to improve the quality of their manuscript.

1.       A few typos, please leave space between words and numbers of references (line 52, 64,69, 70,71,86, 134, 227, 239).

2.      In section 2.1, please add the reagents and the instruments here and not in the supporting info,

3.      In figure 1 the letters are too small to read, please use bigger size fonts.

4.      In the introduction the authors need to mention that one of the biggest disadvantages of this method is that the visual detection of AFB1 with particularly high sensitivity still remains to be a big challenge that limits their potential applications in the rapid analysis of real samples. The authors need to clearly state how they overcome this problem with what they propose.

5.      In the introduction the authors mention other test methods for AFB1 detection and need to explain there as well how their method is better that them with respect to cost, long pre-treatment of the samples tested, long time of response, may steps for the preparation and testing, many reagents used etc.

Author Response

Comments 1: A few typos, please leave space between words and numbers of references (line 52, 64,69, 70,71,86, 134, 227, 239).

Response 1: Thanks for pointing it out. We have corrected the mistakes.

Comments 2: In section 2.1, please add the reagents and the instruments here and not in the supporting info.

Response 2: Thanks for your comment. We added the reagents and the instruments in Section 2.1.

Comments 3: In figure 1 the letters are too small to read, please use bigger size fonts.

Response 3 Thanks for your comment. We replaced figure 1 according to your request for better reading experience. Please see as the following:

Comments 4: In the introduction the authors need to mention that one of the biggest disadvantages of this method is that the visual detection of AFB1 with particularly high sensitivity still remains to be a big challenge that limits their potential applications in the rapid analysis of real samples. The authors need to clearly state how they overcome this problem with what they propose.

Response 3 Thanks for your comment. We agreed that the low sensitivity of existing AFB1 assays was due to the interfering effects of real complex matrices. The isothermal amplification methods can improve assay specificity and sensitivity because they can specifically recognize a single copy of the target and amplify the signal by several orders of magnitude. In the introduction, we described in detail why we used RCA as an amplification method. Please see the line 47-54. At the same time, we realized that the RCA nucleic acid amplification method could not directly recognize a small molecule like AFB1. Therefore, our previous work established the conversion of AFB1 antibody-antigen immunorecognition to nucleic acid recognition with the help of the biotin-streptavidin system, which is described in detail in the introduction, lines 63-66. Overall, we redescribed how we achieved improved detection sensitivity in complex matrices. Please see page 2, line 72-94, as the following.

As a class I carcinogen [13], AFB1 is transferred from the environment to food through a chain of irrigation water-field soil-peanut [14]. The current immunological test methods for AFB1 detection are based on colorimetry [15], electrochemiluminescence [16], electrochemistry [17], lateral flow fluorescent strip [18], fluorescence resonance energy transfer [19], and photoelectrochemistry [20]. Nevertheless, these above methods lacked sensitivity to the complex matrice effects. Harnessing the simplicity and specificity of the RCA nucleic acid detection method, we proposed an RCA-POCT method to combine the streptavidin-biotin system and RCA to boost sensitivity in AFB1 detection in complicated matrix. We fabricated a biotinylated antibody-streptavidin-biotinylated single-stranded DNA (biotin-ssDNA) sandwich to initiate RCA for AFB1 detection in real samples. The above-mentioned sandwich complexes enabled simultaneous immune recognition of antigens and signal transformation. Then, we used RCA to amplify biotin-ssDNA to a long-chain ssDNA expediently. The long-chain ssDNA was partially complementary to plenty of fluorescently labeled signal probes, causing strong fluorescence signal production. The fluorescence intensity was recorded by a microplate reader. After optimization, the irrigation water, field soil, and peanut were used to evaluate the linear range, limit of detection, recovery, specificity, repeatability, and reproducibility. Validation was conducted by comparing results via RCA-POCT to those via the HPLC method. This RCA-POCT with high sensitivity and selectivity promises potential in environmental monitoring and food safety.

Comments 5: In the introduction the authors mention other test methods for AFB1 detection and need to explain there as well how their method is better that them with respect to cost, long pre-treatment of the samples tested, long time of response, may steps for the preparation and testing, many reagents used etc.

Response 5: Thanks for your comment. The advantageous points of the RCA-POCT method are 1. utilizing the specificity of RCA isothermal amplification and signal amplification to improve the sensitivity and linear range of the assay; 2. solving the interference effect of the complex matrix; and 3. applying the antibody with better robustness than the aptamer. We compared mainly in terms of utility and sensitivity, but we recognized that the reviewer's mention of comparing cost/reaction time/preprocessing time/assay steps was a good suggestion. Therefore, we compared this RCA-POCT method with other methods in Section 3. Please see page 9, lines 294-301.

Reviewer 2 Report (New Reviewer)

Comments and Suggestions for Authors

Pay attention to the space between words and [refs 5to 8, 23 to 28]

L 22-23: The construction "compared to the" is not suitable. Maybe use just "of"

L 32: 3 LD, 2 LR and 3 MATRIXES? EXPLAIN WHY only 2 LR

L 34: The right would be: 5 x 10-6 to 5,5 x 10-5 for irrigation water, and 5 x 10-5 to 5 ng/mL for field soil and peanut samples.

L50: Suggestion: "Therefore, RCA...biological assay.... analytical methods. For instance, RCA can be easily...and lateral flow assay [8]"

L 58: "... aptamer affinity is strongly ..."

L 222: "field soil" INSTEAD "peanut soil"

L 241: DID YOU MEAN "higher specificity in irrigation water than field soil and peanut?"

Improve the figures quality!

Comments on the Quality of English Language

The english is good.

Author Response

Comments 1: Pay attention to the space between words and [refs 5to 8, 23 to 28]

Response 1: Thanks for pointing it out. We have corrected the mistakes.

Comments 2: L 22-23: The construction "compared to the" is not suitable. Maybe use just "of"

Response 2: We appreciated your advice and corrected the sentence.

Comments 3: L 32: 3 LD, 2 LR and 3 MATRIXES? EXPLAIN WHY only 2 LR

Response 3 Thanks for your comment. We used this method to detect AFB1 in 3 matrixes. There were 3 LD and 3 LR in 3 matrixes. Please see the following. As you can see, the LR in field soil and peanut samples are the same.

Under optimal conditions, we recorded the sensitive detection limits for aflatoxin B1 (AFB1) of 1.94, 16.3, and 37.7 fg/mL (femtogram per microliter), and wide linear ranges with 5 × 10-6 to 5, 5 × 10-5 to 5, and 5 × 10-5 to 5 ng/mL in the irrigation water, field soil, and peanut samples, respectively.

Comments 4: L 34: The right would be: 5 x 10-6 to 5,5 x 10-5 for irrigation water, and 5 x 10-5 to 5 ng/mL for field soil and peanut samples.

Response 3 Thanks for your comment.

Comments 5: L50: Suggestion: "Therefore, RCA...biological assay.... analytical methods. For instance, RCA can be easily...and lateral flow assay [8]"

Response 5: Thanks for your comment. We rewrote the sentence according to your suggestion.

Comments 6: L 58: "... aptamer affinity is strongly ..."

Response 6: Thanks for your suggestion. We corrected the sentence.

Comments 7: L 222: "field soil" INSTEAD "peanut soil"

Response 7: Thanks for pointing out. We corrected the word.

Comments 8: L 241: DID YOU MEAN "higher specificity in irrigation water than field soil and peanut?"

Response 8: Thanks for your comment. We evaluated this method’s specificity in real samples, not only higher specificity in one kind matrix.

This manuscript is a resubmission of an earlier submission. The following is a list of the peer review reports and author responses from that submission.

Round 1

Reviewer 1 Report

Comments and Suggestions for Authors

The authors have proposed an original use of amplification technology for sensitive detection of low molecular weight toxicants based on the integration of immunodetection and rolling circle amplification. Experimental design of the study and presented results accord to demands of the Foods journal. However, some revisions/justifications will be reasonable for improvement of the prepared manuscript:

1. The detected analyte should be clearly indicated as «mycotoxin aflatoxin B1» in the title, whereas its specification as a small molecule is redundant for the title.

2. Line 45. Why «environmental monitor»? «Monitoring» will be more correct tern here.

3. Lines 55-57. The main application of isothermal amplification techniques is revealing of specific oligonucleotides sequences to identify viral or microbial pathogens, as well as other living organisms. The integration of these techniques with the detection of target analytes of other chemical nature is an important and interesting, but only minor part in the application of these techniques.

4. Line 60. The addressing to some reviews (instead of local experimental papers) will be preferable here for overall estimation of the existing investigations.

5. Lines 63-64. Why macromolecules are considered as an alternative to RNA, DNA and proteins?

6. The use of amplification techniques for compounds that are not nucleic acids needs in justification of initial processes providing selective recognition of these compounds. This basic justification with list of the used approaches (immune recognition, aptamers' use, etc.) should be given in the Introduction to indicate better the place of the authors' development among similar works. The actual text is limited by the description of own authors' experience (see lines 68-72) that is not enough to demonstrate the current state-of-the-art.

7. The authors have described in the Introduction only their development for AFB1 detection, and give its comparison with alternate techniques only in the final part of the manuscript as Table 2 with parameters of previous methods. Such presentation remains unclear reasons for the authors' development. Some integrated comments in the Introduction about variety of existing techniques and demands to new alternate developments will be useful.

8. Have the authors data about cross-reactivity of the assay (or cross-reactivity of the antibodies used in it) to aflatoxin M1, a widespread metabolite of aflatoxin B1?

9. References to prototype publications describing samples preparation should be added to the Section 2.7 to be sure that the chosen protocols provide complete extraction of AFB1.

10. The text started from line 163 is not the assay principle. Please create a separate section for it.

11. As follows from Fig. 4A, the assay demonstrates invert dependence of detected signal from the AFB1 concentration. It is a predictable situation for competitive immunoassay techniques. However, it this situation the reasons to reach maximal fluorescence intensities in the course of optimization studies become to be disputable. Potentially competitive assay can have similar LOD and working ranges for its conditions giving different levels of the top plateau of the calibration curve. By this way, the assay could have, for example, some shorter RCA reaction time (see Fig. 3 E) with improved productivity and stored analytical parameters. However, the optimization experiments are considered in the manuscript only as maximizing fluorescent signal in the absence of the analyte. So comments to the optimization part of the study should be re-written to integrate demands of maximal fluorescence intensity and minimal LOD.

12. Procedure for LOD determination is clearly described in the Section 2.5, but its experimental realization is not visualized in the manuscript. The decisions about LODs are not clear from Fig 4A, Fig. S4, Fig. S5 and needs separate consideration. Moreover, the given LOD values for different matrixes seem sometimes mixed and needs additional careful checking.

13. The alternate assays are very poorly described in the first column of the Table 2. Please indicate clearly: (i) AFB1-recognising molecules (antibody, aptamer, etc.); (ii) tools used to form the detected complexes (microplate, chip, dispersed nanocarriers, homogeneous solutions, etc.); (iii) ways for signal detection (colorimetry, fluorimetry, amperometry, etc.). Representatives of traditional immunoassays such as ELISA, LFIA, will be also useful in the Table 2 to illustrate the techniques that are really implemented in practice. The place of instrumental techniques (such as chromatography) should be also briefly characterized in the Discussion or in the Introduction.

14. The row of examples collected in the Table 2 provides well coverage of the LOD interval from 94 fg/mL to 35 pg/mL, but several original developments with low LODs were loss:

DOI  10.1016/j.foodchem.2022.134212 with LOD 100 fg/mL;

DOI 10.1016/j.snb.2014.05.012 with LOD 50 fg/mL;

DOI 10.1016/j.foodchem.2023.136176 with LOD 37.34 fg/mL;

DOI 10.1016/j.foodchem.2023.135856 with LOD 34 fg/mL;

DOI 10.1016/j.talanta.2021.122772 with LOD 33 fg/mL;

DOI 10.1016/j.snb.2021.131238 with LOD 32 fg/mL;

DOI 10.1016/j.bios.2014.01.045 with LOD 25 fg/mL;

DOI 10.1016/j.electacta.2022.139912 with LOD 9.55 fg/mL;

DOI 10.1016/j.saa.2022.121535 with LOD 8.38 fg/mL;

DOI 10.1021/acs.analchem.9b04822 with LOD 5.07 fg/mL;

DOI 10.1039/d2ay01134d with LOD 4.8 fg/mL;

DOI 10.1039/d2ay01682f with LOD 2.84 fg/mL;

DOI 10.1016/j.snb.2021.129528 with LOD 1 fg/mL;

DOI 10.1021/acsami.1c04751 with LOD 0.54 fg/mL;

DOI 10.1016/j.snb.2022.132619 with LOD 0.34 fg/mL

Probably, it would be reasonable to consider some specific group of assays/sensors for AFB1 with clear and visible indication of the reached improvements in comparison with earlier works.

15. 36 tested samples are not visible from Fig. 4C. Please give raw data of their HPLC and RCA testing in the Supplement. Why all samples demonstrated only 6 concentration values in the HPLC studies? Were they artificially spiked? Did any sample demonstrated presence of AFB1 before its spiking?

Reviewer 2 Report

Comments and Suggestions for Authors

Please see my comments in the attachment.

Reviewer 3 Report

Comments and Suggestions for Authors

The authors developed Rolling circle amplification platform method to able detect low concentration of aflatoxin in the environment and food. I consider the topic original and relevant in the field. It addresses a specific gap and the sensitive of aflatoxin detection is higher than other published paper. The authors should consider control groups in each experiment setting. The conclusions consistent with the evidence and arguments presented and addressed the main question posed. The references are appropriate.

Author have done good work, But this paper misses discussion part. The main text of manuscript is only 2600 Words.  It is not enough for a research article. Please use MDPI template  Please include discussion section so that the main text length will be enough for 4000 words. Main text should be Palatino Linotype font 10 Table  and figure legend should be font 9.  All of figure panel  A,B,C should be replace by a,b,c.  Reference should be font 9.   Abstract ELISA and RCA please use full words before the abbreviation appears first time Please correct the typo in Table 2 at bottom line of table 2 RCA-fluorescence please replace a References for ' this work?'   Line 253- 256 'For intra-assay accuracy, recoveries revealed 96.7%-102.5%, 92.5%-99.4%, and 99.0%-103.6% in irrigation water, field soil, and peanut samples, respectively. For 255 inter-assay precision, recoveries ranged from 96.6%-102.7%, 96.8%-102.6%, '   And is recovery rate over 100% reasonable?

Comments on the Quality of English Language

language is good.

Reviewer 4 Report

Comments and Suggestions for Authors

The current study is quite interesting and well organized, however, please address the listed points:

1. In the abstract define the complete meaning of each abbreviated word at first time mentioned.

2. L 39: revise the scientific unit? fg??

3. Write the reference of Table 1.

4. Most of the methods are confusing since they didn't include the original reference!

5. It is very important to add the models of instruments used.

6. L 132: Write in details the HPLC method, the column and detector used.

7. Table 2, change this work to " The current study".

8. Revise the x axis of fig. 4 B and C, add the specifications of axis x of 4 B.

9. Discuss your findings in more details with relevant and new reports.

10. I think the title is general and could be edited to include the actual sample name, not all food or environment

Comments on the Quality of English Language

Could be improved